# A Comparative Study of Intravital CT and Autopsy Findings in Fatal Traumatic Injuries

**DOI:** 10.3390/healthcare10081465

**Published:** 2022-08-04

**Authors:** Roman Kuruc, Andrea Szórádová, Ján Šikuta, Ľubomír Mikuláš, Jozef Šidlo

**Affiliations:** 1Institute of Forensic Medicine, Faculty of Medicine, Comenius University, Sasinkova 4, 81108 Bratislava, Slovakia; 2Medico-Legal Department, Health Care Surveillance Authority, Antolská 11, 85107 Bratislava, Slovakia

**Keywords:** trauma, injury, autopsy, computed tomography, forensic medicine

## Abstract

*Objectives:* Traumatic injuries are one of the severe health problems of our time. In the 21st Century, approximately 4.5 million people worldwide die each year due to trauma. Computed tomography (CT) is widely used to diagnose injuries and offers information on the specific location and extent of organ and tissue damage. In cases of severe trauma, whole-body CT is increasingly used as a standard diagnostic technique. An autopsy is the final diagnostic examination and is still considered the gold standard in diagnostic methods in medicine. The aim of the study was to assess the reliability and accuracy of CT scan results, as well as limits in detecting trauma for forensic purposes. It aims to compare traumatic findings in the antemortem CT results to those observed at autopsy. *Materials and Methods:* We conducted a retrospective–prospective study involving 510 deaths due to trauma. We compared selected traumatic changes in the antemortem CT scan results with the autopsy findings. We obtained data with a detailed analysis of autopsy protocols, photographic documentation from the autopsies, and the interpretation of CT scans from medical documentation. In cases of discrepancies in the findings, we borrowed CT scans, which were repeatedly reviewed by clinical radiologists. *Results:* By comparing the findings of selected injuries detected by antemortem CT and autopsy, we found a correlation of findings in 75.3% and a discrepancy of findings in 24.7% in a set of 510 cases. After repeated targeted assessment of CT images by clinical radiologists in cases of discrepancies in the findings, which were detected by autopsy and undescribed by CT, the discrepancy decreased to 17%. *Conclusions:* The results of the study are comparable with data from many studies and professional publications. They show that CT compared to autopsy is a good method for diagnosing gunshot wounds to the head and bone fractures, with a limited diagnosis of cranial base fractures, while an autopsy is better for detecting minor injuries to organs and soft tissues.

## 1. Introduction

Traumatic injuries are one of the severe health problems of our time. In the 21st Century, approximately 4.5 million people worldwide die each year due to trauma. Millions more people who survive the trauma have permanent disabilities and consequences. In 2017, injuries accounted for 8.0% (7.7–8.2) of total deaths, corresponding to 4.48 million (4.33–4.59) deaths. Overall, from 2007 to 2017, there were 20.1 million (18.7–20.8) deaths from unintentional injuries, 15.1 million (14.8–15.4) deaths from transport injuries, and 14.4 million (13.7–14.7) deaths from self-harm and interpersonal violence. Total deaths from injuries increased slightly by 2.3% between 2007 and 2017, and the death rate from injuries decreased by 13.7% (12.2–15.1) to 57.9 deaths (55.9–59.2) per 100,000 inhabitants in 2017 [1].

Computed tomography (CT) is chiefly used to investigate and diagnose traumatic injuries [2]. CT plays an important diagnostic role in current clinical medicine [3]. The great advantage of CT is availability, rapidity, and relatively high diagnostic accuracy. Computed tomography offers specific information on the area and extent of damage to internal organs and tissues [4]. The number of CT examinations is constantly increasing in clinical conditions. In patients with severe trauma, whole-body CT (WBCT), which minimizes missed injuries, is becoming increasingly a standard diagnostic technique [5]. In addition to rapidity, the main advantage of WBCT is the early availability of a diagnosis before a decision on treatment management [4,6,7]. Rapid and accurate recognition of traumatic injuries is extremely important. Without an accurate evidence-based diagnosis, there is no effective treatment.

An autopsy is a specialized medical procedure. It is the ultimate diagnostic test, the gold standard in diagnostic methods, with an assumed 100% sensitivity for finding causes of death and 100% specificity for excluding them. The persistent value of autopsies does not mean that medicine has failed to advance, but rather that every advance brings new challenges and new unknowns [8]. Autopsy continues to play a key role in the development of medical practice and science [9]. According to an extensive study, it was found that in research papers published from 1966 to 1993, an increase in information obtained from autopsies by 220% was recorded [10]. In patients with trauma, the significance of autopsy becomes paramount in cases of so-called “missed” diagnoses. The true incidence of clinically “missed” diagnoses in trauma-related deaths is unknown, as it is estimated that autopsy is conducted in only about 50% of cases [11]. Some physicians believe that diagnosis using modern imaging methods has improved to the point that autopsies are no longer needed. In Western countries, autopsy rates for patients that have died in hospitals have dropped to a record low, while the frequency of major errors in clinical diagnoses has more than doubled over the same period [12]. The results of several studies show that a higher number of autopsies results in improved accuracy of clinical diagnoses.

The study aims to compare antemortem CT scan results with autopsy findings in cases of trauma-related deaths and evaluate the reliability and accuracy of CT results. Autopsy was assigned as the gold standard of diagnosis.

## 2. Materials and Methods

### 2.1. Ethical Approval

This retrospective–prospective study was approved by our Institutional Ethics, Arbitration and Disciplinary Committee of the Faculty of Medicine, Comenius University, in Bratislava, Slovakia.

### 2.2. Study Design

The study was conducted as an analysis of autopsy protocols from the autopsies performed at the Medico-legal Department of the Health Care Surveillance Authority in Bratislava over 9 years (2010–2018). In all cases, a standard autopsy was performed with interpretation and photographic documentation of the injuries and histopathological examination. The study included traumatic deaths in which antemortem CT was performed followed by autopsy. Selected findings of injuries found at autopsy and the last antemortem CT were compared and evaluated as one diagnosis. Furthermore, the mechanism and cause of death (according to ICD), gender, age, and length of survival were evaluated. Injury data were obtained by detailed analysis of autopsy protocols, photographic documentation from autopsies, medical documentation, and results of CT examinations. All antemortem CT scans were acquired on spiral or multidetector CT scanners. In all cases of discrepancies in autopsy and CT findings, CT records were borrowed and reviewed by experienced clinical certified radiologists (>10 years work experience each) focusing on autopsy-detected traumatic changes. The image revision was performed blindly, and radiologists made their decisions independently.

### 2.3. Data Analysis

Analysis was performed to assess the correlation between the autopsy and CT scan results. Sensitivity (percentage of those with injury that have injury detected on CT scan) and positive predictive value (percentage of positive CT results that have injury) were defined for the CT scan based on the autopsy results.

### 2.4. Cases

From 2010 to 2018, 6861 autopsies at the Medico-legal Department of the Health Care Surveillance Authority in Bratislava were performed. Furthermore, 1654 deaths in medical facilities were reported, which accounts for 24.1% of all autopsies. A set of 1654 autopsy protocols was thoroughly analyzed to find suitable cases for the study.

Cases were included in the study if an autopsy was conducted within 48 h after death, a CT scan was performed within 48 h before death and in cases of fractures within 96 h before death, and there was no surgical intervention. Another criterion was sufficient radiologic and morphological interpretation of the traumatic findings necessary for comparison.

We grouped cases into five groups on the basis of the external cause of death (death mechanism) and whether it was an isolated injury to one or more regions of the body: isolated blunt head injuries, transport accidents and falls from a height, gunshot wounds to the head, hangings, and sharp object injuries.

## 3. Results

Five hundred and ten cases were identified that met the study criteria. There were 373 males (73.1%) and 137 females (26.9%) in the set. Patients ranged in age from 9 to 88 years, and the mean age was 57 years. The age group 51–60 years (25.7%) had the largest representation, then 61–70 years (21.5%). The average length of hospitalization in a medical facility was 7.2 days. More than a quarter of cases (26.8%) were deaths that occurred within 24 h of admission to the medical facility and more than half of the cases (52.4%) of deaths during the first 3 days of hospitalization. The most common cause of death was brain death and traumatic hemorrhagic shock.

The final set was divided into groups and the external cause of death (death mechanism) was determined as a basic criterion. At the same time, it was taken into account which body areas were traumatically damaged and whether it was an isolated injury to one or more regions of the body. Accordingly, the set of 510 deaths was divided into the following groups: isolated blunt head injuries (308 cases), transport accidents (133 cases), falls from a height (42 cases), gunshot wounds to the head (11 cases), hangings (10 cases), and sharp object injuries (6 cases).

### 3.1. Isolated Blunt Head Injuries

A total of 308 cases were included in this group (60.4% of the final set). The most common cause of death was brain death. There were 225 cases of injuries with skull fractures (210 fractures of the skull and/or base, 15 fractures of the facial bones). There were 122 cases of fractures of the cranial vault and the base, in 56 cases only a fracture of the cranial base, and in 32 cases only a fracture of the cranial vault. Skull fractures were combined in 94.8% of cases with extracerebral post-traumatic hemorrhage (subdural, subarachnoid, and epidural) and 85.4% with brain contusion. Intraventricular hemorrhage was detected in 43 cases (20.5%). Facial bone fractures were combined with brain contusion (66.7%) and subdural hemorrhage (33.3%). There were 83 cases of injuries without skull fractures. The most common cases were subdural hemorrhage (48 cases), brain contusion (21), subarachnoid hemorrhage (11), and epidural hemorrhage (3 cases). In all cases, only a CT scan of the head was performed.

The comparison of injuries detected by antemortem CT and autopsy included skull fractures, extracerebral post-traumatic hemorrhage, brain contusion, and cerebral ventricular hemorrhage. Brain contusion due to the dynamics of the development of traumatic changes included only the cases where the patient died within 24 h after the last CT examination (64% cases within 6 h, 27% within 12 h, 6% within 18 h, and 3% within 24 h), and autopsy revealed deposits of brain contusion larger than 0.5 cm in diameter. Injuries to the soft skull fascia and temporal muscles were not included in the comparison due to the absence of their interpretation in many clinical CT examinations.

The comparison of CT and autopsy findings in a group of 308 cases of isolated blunt craniocerebral head injuries revealed a correlation in 235 cases (76.3%) and a discrepancy among findings in 73 cases (23.7%). Of the 73 cases of discrepancies, 61 were (83.6%) not described by CT, but detected by autopsy (false negative). In the remaining 12 cases (13.4%), CT examinations described injuries that were not confirmed at autopsy (false positive). For head injuries, the sensitivity of a CT scan was 79.4%, and the positive predictive value was 95.1%.

In 61 cases of injury discrepancies that CT did not describe, there were 33 cranial vault and/or base fractures, representing 15.7% of fractures undetected by CT examination of all cranial vault and base fractures. The most common were fractures of the head bone in the posterior cranial fossa (19 cases; 57.6%), followed by fractures of the parietal and temporal bones passing to the cranial base (8 cases; 24.2%) and fractures of the orbital roof (6 cases; 18, 2%). There were 10 cases of brain contusion undescribed by CT (contusions detected by autopsy were found at the base of the frontal and/or temporal lobes); subarachnoid hemorrhage was in 9 cases; subdural hemorrhage was in 4 cases (subarachnoid and subdural hemorrhage detected by autopsy was most common in the base of the brain), epidural hemorrhage in 4 cases, and intraventricular hemorrhage in 1 case. False negative cases of CT scans for each type of injury are summarized in Figure 1A. In a group of 12 cases of discrepancies in which the traumatic findings described by CT were not detected at autopsy, there were 6 fractures of the cranial vault and base (parietal bone, petrous pyramid), 3 contusions of the base of the brain, and 3 cases of epidural hemorrhage. False positive cases of CT scans for each type of injury are summarized in Figure 1B.

Repeated assessment of CT images in 61 cases of discrepancies diagnosed 24 injuries: 17 fractures of the cranial vault and/or base, 2 brain contusions, 2 subarachnoid hemorrhages, 2 epidural, and 1 subdural hemorrhage. A discrepancy of findings in the craniocerebral head injury group dropped from 23.7% to 15.3%. CT examination left 37 undiagnosed injuries. False negative cases after repeated assessment of CT scans for each type of injury are summarized in Figure 2.

### 3.2. Transport Accidents and Falls from a Height

There were 133 cases of transport accidents (26.1% of the final set) and 42 cases of fall from a height (8.2% of the final set) in this group. Whole-body CT examination was performed in 143 cases (81.7%); in the remaining cases, a CT scan of the trunk and pelvis was performed. Head injuries were found in 126 cases, cervical spine injuries in 5 cases, chest injuries in 117 cases, abdominal injuries in 98 cases, and pelvic injuries in 28 cases. The most common cause of death was brain death and traumatic hemorrhagic shock.

The correlation of autopsy and antemortem CT involved fractures (skull, spine, collarbones, ribs, sternum, pelvis), extracerebral hemorrhages (epidural, subdural, subarachnoid), brain injuries (contusions, intraventricular hemorrhages), and injuries to internal organs in the trunk. Extremity injuries and bleeding into the thoracic and abdominal cavities, subcutaneous hematoma, or ruptures were not included in the comparison.

The comparison of CT and autopsy findings in a group of 133 transport accidents and 42 falls from a height revealed a correlation in 128 cases (73.1%) and a discrepancy in 47 cases (26.9%). In more than half of the cases of discrepancies (35), there was a discrepancy between two injuries, in 1 case three injuries, and 11 cases one injury. The total number of discrepancies was 84 injuries (diagnoses). In this group, the sensitivity of a CT scan for injuries was 75.7%, and the positive predictive value was 95.5%.

The most frequent CT-undescribed, but autopsy-confirmed injuries were fractures (36 cases). The most common were rib fractures (18), then skull base fractures (11), pelvic fractures (5 cases of sacroiliac joint injury), fracture of the sternum (1), and cervical spine fracture (1 case; CT did not reveal a lesion of the atlanto-occipital junction). There were also 11 cases of CT-undescribed intracranial injuries—contusions of the base of the brain (4), subdural hemorrhage (4), subarachnoid hemorrhage (2), and epidural hemorrhage (1). Of internal organ injuries found at autopsy, CT did not describe contusions and/or ruptures of the lungs (11), liver ruptures (6), spleen ruptures (4), kidney ruptures (2), thoracic aortic rupture (2), nor rupture of the heart and pericardium (1). False negative cases of CT scans for each type of injury are summarized in Figure 3A. A group of cases in which the findings were described by CT examination and not detected at autopsy (false positive CT scans) included skull fractures (2), collarbone fracture (1), sternal fracture (1), epidural hemorrhage (1), thoracic aortic rupture (1), pulmonary contusion (2), liver ruptures (2), and spleen rupture (1). False positive cases of CT scans for each type of injury are summarized in Figure 3B.

Repeated evaluation of CT scans revealed rib fractures (7), skull base fractures (3), subdural hemorrhage (1), pulmonary contusion (4), liver ruptures (2), spleen rupture (1), and kidney rupture (1). Discrepancy dropped from 26.9% to 20.8%.

Several fractures of the skull base and ribs, then subarachnoid and epidural hemorrhages, base brain contusions, fracture of the cervical spine and sternum, injuries to the sacroiliac joint of the pelvis, and injuries to internal organs, including a rupture of the thoracic aorta, heart, and pericardium, remained undetected by CT scans. False negative cases after repeated assessment of CT scans for each type of injury are summarized in Figure 4.

### 3.3. Gunshot Wounds to the Head

A total of 11 cases were included in this group (2.2% of the final set). In 9 cases, a short bullet firearm was used and in 2 cases, a slaughter pistol. There were 6 perforating wounds and 5 penetrating wounds to the head. The cause of death was brain death. A CT scan of the head was performed in all cases.

The following injuries were compared: skull fractures, extracerebral hemorrhage (epidural, subdural, subarachnoid), intraventricular hemorrhage, and brain contusion. Furthermore, the description of the course of the wound track and the location of the finding of the bullet in the cranial cavity was compared. Injuries to the soft skull fascia and temporal muscles were not included in the findings reviewed due to the absence of a description of these injuries in almost all CT scans.

After comparing injuries found at autopsy and described by CT examination, a correlation was found in 10 cases (90.9%) and discrepancy in 1 case (9.1%). It was the course of the wound track of the perforating wound to the head that was incorrectly described by CT examination in the opposite direction as it was verified at autopsy. By repeatedly assessing the CT images, despite making a 3D skull model, it was impossible to identify the wound track’s correct orientation. The sensitivity of a CT scan for injuries was 90.1%, and the positive predictive value of a CT scan for injuries was 100%.

### 3.4. Hangings

In a group of hangings, 10 cases were evaluated (2.0% of the final group). In 5 cases, the strangulating instrument was a car towing rope, in 3 a trouser belt, and in 2 a narrower cord. The cause of death was a hypoxic brain injury. A CT scan of the head and neck was performed in all cases.

Comparison of radiological CT and morphological autopsy findings included injury changes related to hanging—fractures of the hyoid bone, fractures of the laryngeal cartilage, and bleeding into the soft tissues of the neck.

Comparison of CT and autopsy findings found a correlation in 7 cases (70%) and discrepancy in 3 cases (30%). It was a discrepancy when the injuries detected by autopsy were not described by CT scan (false negative CT scans). In two cases, a fracture of the greater horn of the hyoid bone was not described, while in one case, this was a fracture of the superior horn of the thyroid cartilage with bleeding into the adjacent tissues. The sensitivity of a CT scan for injuries was 70%, and the positive predictive value of a CT scan for injuries was 100%.

Repeated assessment of CT records of the three cases of discrepancy revealed in one case a fracture of the greater horn of the hyoid bone, thus reducing the discrepancy from 30% to 20%.

### 3.5. Sharp Object Injuries

This group included six cases of death (1.2% of the final set) that occurred due to the action of a stabbing tool (knife). In 5 cases, a stab wound of the trunk was found penetrating the thoracic (4) and abdominal cavities (1) with injury to internal organs (heart, lungs, liver). In one case, it was a stab wound to the head penetrating through the left orbit into the cranial cavity, with a brain injury. The cause of death was a traumatic hemorrhagic shock and brain death in the case of a stab wound to the head.

The monitored parameters included injuries to internal organs (brain, heart, lungs, liver), orbit fracture, and bleeding into the thoracic and abdominal cavities. In the case of a stab wound to the head, a CT scan of the head was performed and, in other cases, a CT scan of the trunk. After comparing CT and autopsy morphological findings, a correlation of findings was found in 4 cases (66.7%) and discrepancies in 2 cases (33.3%). In both cases, a stab wound to the lungs within a stab wound to the back was not described by CT examination. In the first case, it was a complete stab wound to the lower lobe of the left lung and, in the second case, a stab wound to the right lung of 2.5 cm in depth. The sensitivity of a CT scan for injuries was 66.7%, and the positive predictive value of a CT scan for injuries was 100%. Repeated CT assessment revealed a perforation of the lower lobe of the left lung. Even repeated CT assessment failed to diagnose a stab wound to the lungs of about 2.5 cm deep in the second case. A discrepancy of the findings decreased from 33.3% to 16.7%.

## 4. Discussion

In this study, the autopsy findings and the last antemortem CT scans were compared in selected traumatic findings in the head, cervical spine, trunk, and pelvic injuries. Autopsy as the gold standard was assigned for statistical evaluation.

After evaluating the results in a set of 510 deaths, a correlation was found between autopsy and CT findings in 384 cases, which accounts for 75.3%. The total number of false negative CT scans in comparison with the autopsy is summarized in Figure 5. There was a discrepancy between the findings in 126 cases, which accounts for 24.7%. The total number of false positive CT scans in comparison with the autopsy is summarized in Figure 6.

After repeated targeted reviewing of CT images by clinical radiologists, the discrepancy rate decreased to 17%.

The outcomes of our study are consistent with the conclusions of the study by the authors from the University of Pittsburgh who compared antemortem radiologic diagnoses with autopsy findings. They reviewed 729 diagnoses, and 27.6% of them were determined to be discrepant from the corresponding radiologic and autopsy diagnoses. The results of their study showed that autopsies could help radiologists sharpen their skills in interpreting radiologic findings and can perhaps serve as quality control for radiology. They also suggested that radiology can partially serve as quality control for autopsies [13].

In a group of blunt head injuries, where 308 cases were reviewed, a correlation between CT and autopsy findings was found in 76.3%. The results correspond to the conclusions of the study of German authors, who, in a set of 86 cases of cranio-cerebral trauma, found a correlation in 72% of cases [14]. Of the total 210 fractures of the cranial vault and/or base, 33 fractures were not described, accounting for 15.7% of unnoticed fractures by CT compared to autopsy. This corresponds to the Indian authors of a comparative study on the sensitivity and specificity of CT examination in the detection of skull fractures, in which 60 cases were reviewed with the conclusion of 14.6% of unnoticed fractures by CT [15]. The same authors in another study reviewed 42 cases of head injuries by autopsy and found fractures in 28 cases and, by CT examination, fractures in 25 cases, which is a correlation of 89.3% [16].

Regarding skull fractures, CT did not detect mainly fractures of the occipital bone in the posterior cranial fossa, followed by fractures of the middle cranial fossa, parietal and temporal bones, and the orbital roof. These results correspond to a retrospective study conducted at the Department of Forensic Medicine, the University of Copenhagen, in which, of 56 analyzed cases of fractures of the cranial base, CT examination failed to diagnose 22% of fractures, mostly fractures of the posterior cranial fossa, then the middle fossae, and at least the anterior cranial fossa [17]. According to another study, a detailed analysis of 14 cases of skull fractures caused by blunt force was performed. The results showed a good diagnostic correlation regarding fractures localized in the posterior fossa. In contrast, the finding of fractures in the middle and particularly anterior fossa was difficult to assess [18]. A study by Swiss authors from the Institute of Forensic Medicine of the Universities of Bern and Zurich showed a high sensitivity in detecting orbital roof fractures, but significantly less specificity than an autopsy. Autopsy, therefore, remains the gold standard when examining orbital roof fractures. The study also showed that CT is also a good tool for detecting retrobulbar hemorrhage as one of the “blind spots” in autopsy [19]. A study by the authors from France comparing postmortem CT findings with autopsy findings in a set of 236 cadavers showed a correlation in both fractures of the skull and cranial base, which is a different conclusion from previous studies and our findings. However, according to this study, discrepancies were found in intracranial injuries, especially hemorrhage, as well as injuries to internal organs, when a higher number of diagnoses was found at autopsy [20]. The correlation between radiologic and autopsy findings, in up to 80% of cases, was also described by the Korean authors in their study. They found that CT provides reliable diagnoses, especially in cases of head injuries. However, it should be noted that this study includes an analysis of only five deaths [21].

A high degree of correlation in our study was found in brain contusion. This is consistent with the results of a study by German authors comparing pathological findings between autopsy and antemortem CT after traumatic brain injury. The results of the comparison revealed a high specificity (≥80%) in most of the findings [22]. In a group of traumatic extracerebral hemorrhages, our study found the strongest correlation between CT and autopsy findings in subdural and subarachnoid hemorrhage and the smallest in epidural hemorrhage. These results are in partial correlation with another study by American authors. Their study shows the highest CT scan sensitivity in cases of subdural hemorrhages (66%) and low sensitivity in epidural hemorrhages (33%). The sensitivity of CT scans in subarachnoid hemorrhages was 44%, while in our study, we found a correlation in 83% [23]. In a retrospective study aimed at finding a correlation between postmortem CT (pmCT), MRI, and autopsy, where 40 cases were analyzed, some differences were found compared to our results for subdural and subarachnoid hemorrhage. According to this study, 82% of subdural and 89% of subarachnoid hemorrhages were diagnosed postmortem. A significant difference in the results was in epidural hemorrhage, which was diagnosed in up to 95% of cases, while in our study, we found a correlation in 77% of cases [24]. According to another retrospective study focused on intracranial traumatic changes detected by pmCT and autopsy, in which 33 cases were randomly selected, subdural and subarachnoid hemorrhage were detected in 4 cases by autopsy. PmCT did not reveal subdural or subarachnoid hemorrhage in any of these cases, so this review confirmed that discrepancies between radiologic pmCT and autopsy findings persist [25]. However, it should be noted that according to the study of the authors from Japan, which deals with the issue of reducing the volume of subdural hematoma between the agonal and postmortem periods by comparing CT scans obtained just before death and after death, the postmortem CT image has specific findings that differ from the antemortem CT image [26].

A total of 175 cases were reviewed in the group of deaths due to transport accidents and falls from a height. It was mostly polytrauma. Comparison of CT and autopsy findings revealed a correlation in 73.4% of cases and a discrepancy in 26.6%. Among the most common trunk injuries in which we found discrepancies were rib fractures and lung, liver, and spleen injuries. A higher percentage of correlation is reported in the study of the authors from the University of Pennsylvania in the USA. However, they compared postmortem CT findings with autopsy findings in only 12 cases of trauma-related death and found a correlation in 83% of cases between pmCT and autopsy [27]. A study of the authors from the University of Bern in Switzerland, whose aim was to determine by pm CT the sensitivity and specificity for selected abdominal injuries after major blunt trauma, includes a review of 34 cases. The sensitivity for liver injury in postmortem computed tomography was 53%. For many cases in which superficial liver lacerations were detected by autopsy, it seemed that they could not be diagnosed by pmCT [28]. Furthermore, in another study, the Dutch authors present that the findings most frequently not detected by pmCT were minor injuries to internal organs, such as superficial liver lacerations [29]. The study of the American authors shows the highest CT scan sensitivity at detecting liver injuries (75%), but only four cases were included in their study. Of all other trunk injuries, CT scans had a sensitivity of less than 50% [23]. Another study evaluated 67 deaths after transport accidents. Postmortem CT examination revealed 994 diagnoses of injuries, which were compared with diagnoses obtained by autopsies. CT examination detected more skeletal injuries and autopsy more injuries to internal organs [30]. Our study shows that CT examination revealed fewer rib fractures than autopsy, especially fractures of the anterior rib sections. In one case, we confirmed that a thoracic aortic rupture diagnosed by CT examination was not confirmed at autopsy; autopsy revealed a rupture of the atrium of the heart. Another study evaluating 20 cases of fatal falls from a height confirmed our findings that falls from a height most often result in chest injuries, especially in the case of rib fractures. Cranial fractures were identified in this study only as the second-most common form of fracture [31]. In pelvic injuries, CT examination did not reveal any sacroiliac joint injuries in five cases, which accounts for 17.8%. This finding correlates with the findings of a study by the German authors, who reviewed 19 cases of pelvic fractures and found that autopsy was superior in detecting sacroiliac joint injuries [32].

In a group of 11 cases of gunshot wounds to the head, we found a discrepancy between CT and autopsy in only 1 case. It was the course of the wound track that could not be identified even by the repeated assessment of CT. Nevertheless, we found that by CT, it is possible to distinguish with high accuracy the entrance wound and exit wound and the course of the wound track and to prove the presence of even tiny “secondary projectiles”. The 3D models of the skull created facilitate better orientation than a classic 2D image. Several professional publications and studies prove that CT is an effective imaging technique in diagnosing and localizing gunshot wounds [33]. Other published papers also show that CT and three-dimensional imaging techniques can be useful tools for evaluating gunshot wounds of the skull. They achieve very good results, in some cases even better than conventional autopsy, in identifying the bullet entrance wound and exit wound, determining the course of the wound track, the extent of brain injuries, and, in the case of a penetrating wound, identifying the bullet and its fragments [34]. Other professional papers dealing with 3D models and biometric reconstruction allow the visualization and determination of the angle of the projectile in all three planes [35]. Some studies demonstrate certain limitations of CT, especially in gunshot wounds to the trunk. In one of the published studies, which examined 13 cases of fatal gunshot wounds, it was found that pmCT was able to determine gunshot wounds and injuries to internal organs. Furthermore, in four cases of multiple gunshot wounds with the crossing of wound tracks, CT led to incorrect determination of the number of gunshot wounds [36].

In a group of 10 hanging cases, a discrepancy was found in 3 cases. In two cases, CT did not reveal a fracture of the greater horn of the hyoid bone and, in one case, a fracture of the superior horn of the thyroid cartilage with bleeding into the adjacent region. A repeated review of CT scans revealed a fracture of the greater horn of the hyoid bone, thus reducing the discrepancy to 20%. A study from the Institute of Forensic Medicine of the University of Bern, analyzing the deaths of nine people who died from hanging or manual strangulation, compares the soft tissue findings of the neck found by pmCT, MRI, and autopsy. Except for one minor bleeding and vocal cord hemorrhage, identical findings were found, which is slightly inconsistent with our findings. The study reports that CT and MRI imaging methods offer great potential for the forensic examination of lesions due to strangulation and suffocation also in clinical forensic medicine [37]. In another study, the authors discussed the possibility of detecting fractures of the laryngeal cartilage and injuries to the surrounding soft tissues of the neck in strangulation by CT examination. In a group of eight patients, they performed pmCT, the results of which they compared with the results of autopsies. In six cases (75%), they found a correlation of fractures, which is in line with our findings. In addition, in two cases, CT revealed cartilage fractures that were not detected at autopsy. Bleeding into the soft tissues of the neck was detected at autopsy in five cases, but only in one case by CT. Therefore, CT is not sufficient to diagnose neck soft tissue injuries [38].

In a group of 6 cases of sharp object injuries, we found a discrepancy of findings between CT and autopsy in 2 cases. It was a stab wound to the lungs. Furthermore, a Brazilian authors’ publication describes a case of a stab wound to a man who was stabbed in the chest and suffered three stab wounds. PmCT scan and autopsy were performed. Comparing the findings, it was found that the autopsy provided better results in the case of external injuries (stab wounds). An initial CT examination did not detect a stab wound to the scapula. Lesions and the course of the stab wound trajectory in the lungs and hemothorax were described by CT and autopsy similarly. CT better described the vascular lesions. According to previous analyses, the difference between conventional autopsies and CT descriptions from stab wounds is less than 20% [39]. According to another study, CT scanning of the chest has limitations, and minor organ injuries may be more difficult to detect. Despite a high sensitivity of CT, the usefulness of diagnosing occult chest injuries remains controversial [40]. Another study reports that the use of chest CT for screening patients with penetrating chest trauma has increased considerably in the last decade, with injuries often affecting the lungs, and the traces of knife injuries may be very subtle and may not be detectable by CT [41].

Our study has some limitations. First, forensic physicians were not blinded to the antemortem CT findings prior to autopsy, and this represents a potential source of bias. Second, the radiologists who performed antemortem CT evaluation were not completely blinded to clinical history. In this study, in all cases of discrepancies in autopsy and CT findings, CT scans were borrowed for repeated assessment. A certain limitation of the study was the interpretation of CT findings, which is related to the experience of radiologists. Therefore, all repeated evaluations were performed by experienced clinical certified radiologists (with more than 10 years’ work experience) with a focus on traumatic changes detected by autopsy. The image revision was performed blindly, and radiologists made their decisions independently.

Nevertheless, this study is representative of normal clinical practice and provides a realistic description of the accuracy and limitations of CT examination in the diagnosis of injuries.

## 5. Conclusions

The authors of the article are forensic physicians, for whom the primary interest is an accurate description of injuries and a reliable determination of the final diagnosis. In forensic medicine, a definitive diagnosis must be established with certainty. This is important not only in determining the cause, but also the mechanism of death. The objective of this study is one of forensic importance. There are only a few studies comparing antemortem CT and autopsy findings in the medical literature to date to our knowledge, so our study stands in an important place among them. The majority of the published studies compare postmortem CT and autopsy findings. The strength of our study is its large sample size. The results of this comparative study are comparable to those in many other studies and professional publications. The results show that CT is a good tool for diagnosing gunshot wounds to the head, bone fractures with a limited diagnosis of fractures of the cranial base and ribs (especially anterior sections of ribs), subdural and intraventricular hemorrhage, and relatively good for liver and spleen injuries. CT also allows investigation of anatomic regions that are not easily visible during autopsy.

On the contrary, an autopsy is better for detecting skull base fractures, rib fractures, subarachnoid hemorrhage, minor injuries to multiple internal organs (brain, heart, lungs, aorta, kidneys), and subcutaneous soft tissue injuries. This study showed that CT examination revealed several, but not all injuries. Oppositely, multiple injuries detected by CT scans were not confirmed at autopsy. This means that CT’s diagnostic potential is limited, CT scans are not yet an adequate detection tool for forensic purposes. Unless the diagnosis of all traumatic changes by CT is not possible, CT examination is as yet recommended as an important additional tool for detecting traumatic injuries and the cause and mechanism of death in the field of forensic medicine. In forensic medicine, specific injuries may be present that are clinically insignificant, but extremely important in terms of the mechanism of injury and manner of death. CT can significantly complement the findings of an autopsy. However, at present, it still cannot be considered as a replacement for traditional autopsy. CT scanning can only be seen as a substitute for an autopsy in a minority of cases. Therefore, it is necessary to maintain conventional autopsy as a basic examination and diagnostic method in forensic practice. The combination of CT scan and autopsy seems to be an ideal tool for diagnosing injuries for forensic purposes. This study pointed out the need to equip forensic workplaces with the possibility of postmortem CT examinations for the sake of the development of forensic science in the field of postmortem diagnostics.

## Figures and Tables

**Figure 1 healthcare-10-01465-f001:**
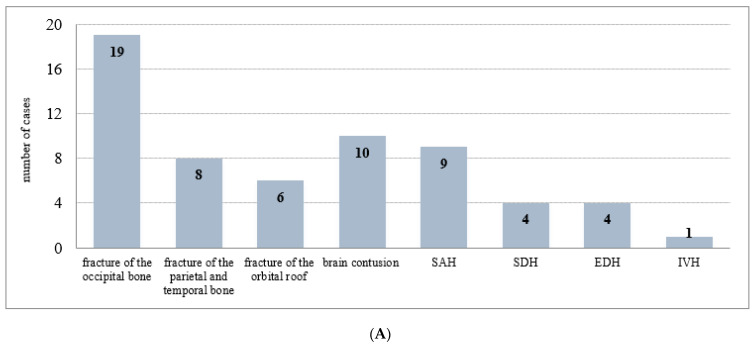
(**A**) The false negative cases of CT scans (61) for each type of injury. SAH—subarachnoid hemorrhage; SDH—subdural hemorrhage; EDH—epidural hemorrhage; IVH—intraventricular hemorrhage. (**B**) The false positive cases of CT scans (12) for each type of injury. EDH—epidural hemorrhage.

**Figure 2 healthcare-10-01465-f002:**
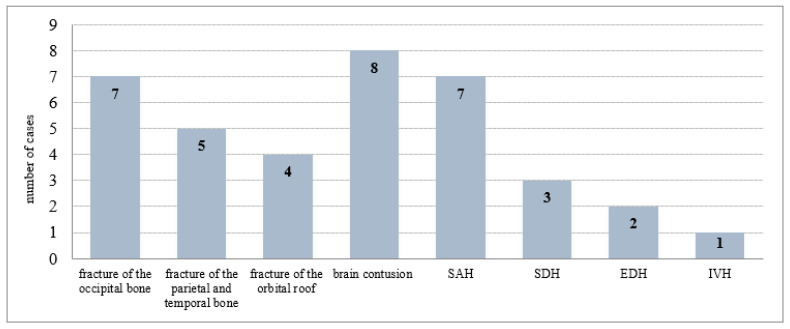
The false negative cases after repeated assessment of CT scans (37) for each type of injury. SAH—subarachnoid hemorrhage; SDH—subdural hemorrhage; EDH—epidural hemorrhage; IVH—intraventricular hemorrhage.

**Figure 3 healthcare-10-01465-f003:**
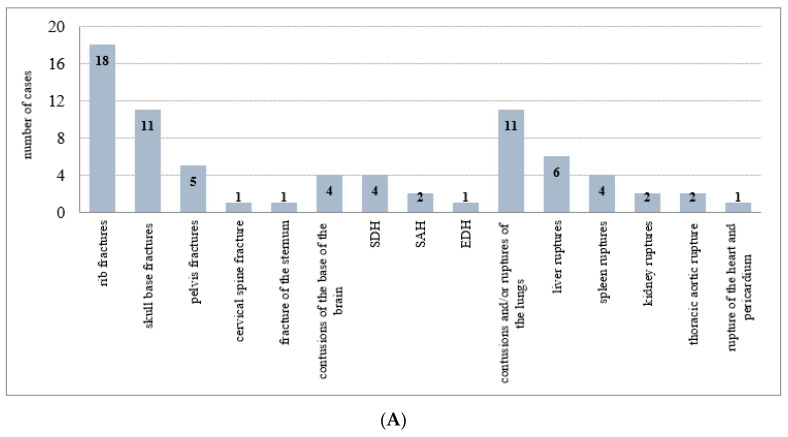
(**A**) The false negative cases of CT scans (73) for each type of injury. SDH—subdural hemorrhage; SAH—subarachnoid hemorrhage; EDH—epidural hemorrhage. (**B**) The false positive cases of CT scans (11) for each type of injury. EDH—epidural hemorrhage.

**Figure 4 healthcare-10-01465-f004:**
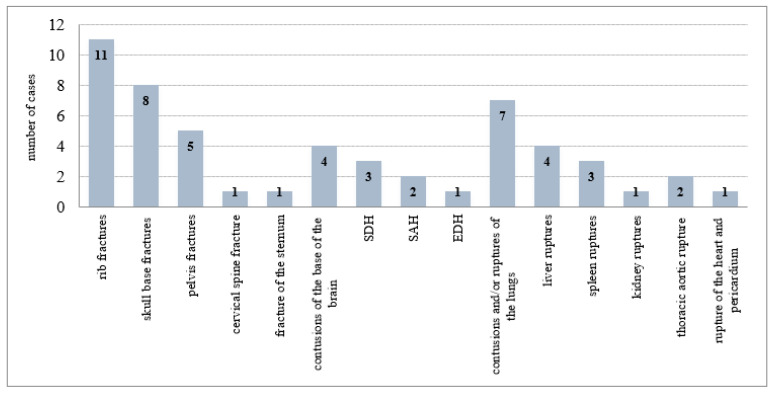
The false negative cases after repeated assessment of CT scans (65) for each type of injury. SDH—subdural hemorrhage; SAH—subarachnoid hemorrhage; EDH—epidural hemorrhage.

**Figure 5 healthcare-10-01465-f005:**
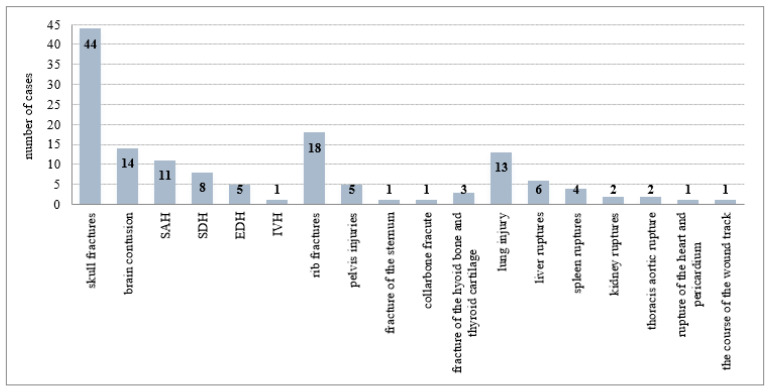
Total number of false negative CT scans in comparison with the autopsy. SAH—subarachnoid hemorrhage; SDH—subdural hemorrhage; EDH—epidural hemorrhage; IVH—intraventricular hemorrhage.

**Figure 6 healthcare-10-01465-f006:**
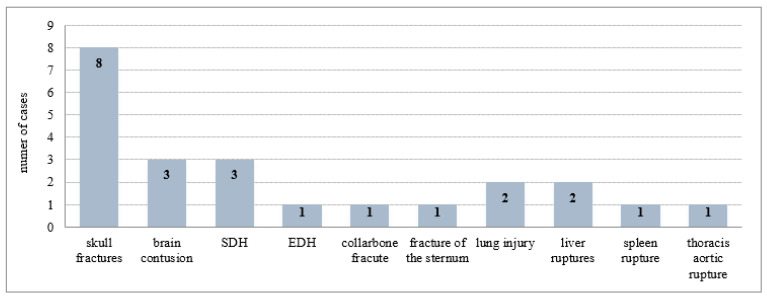
Total number of false positive CT scans in comparison with the autopsy. SDH—subdural hemorrhage; EDH—epidural hemorrhage.

## Data Availability

The datasets generated and/or analyzed during the current study are not publicly available, but are available from the corresponding author upon reasonable request.

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
