# Peer review of "A Comparative Study of Intravital CT and Autopsy Findings in Fatal Traumatic Injuries"

_healthcare, 2022, doi:10.3390/healthcare10081465_

Round 1

Reviewer 1 Report

This paper is a retrospective-prospective study aiming to compare traumatic findings found in antemortem CT and autopsy. The sample size and the evidence emerged from the analysis of the sub-groups are interesting for forensic fields. Particularly, even if the results are mostly comparable with data provided by other studies, the work offers useful information for forensic practitioners describing both the diagnostic limits of CT and autopsy advantages.

The Introduction is well focused on the background of the work, clearly explaining the epidemiological data and the aim of the study.

Material and methods should include the inclusion and exclusion criteria reported in the Results.

Results are both clearly presented and schematically summarized by graphics.

In the Discussion, the reference to statistical evaluation is inappropriate because a statistical analysis has not been performed; the authors reported only the percentage of the cases in which concordance and discrepancy, respectively, were observed.

The authors should make some considerations about the influence of radiologist skill and expertise on CT images analysis. This is important considering that the image revision has not been blindly performed.

The Conclusions are good. However, the Authors should better express the concept at line 458 “Unless the diagnosis by CT is 100% ...”

Author Response

Comments from Reviewer 1

Comment 1: Material and methods should include the inclusion and exclusion criteria reported in the Results.

Response: We agree with this comment. Therefore, we have added the inclusion and exclusion criteria to Material and methods.

Comment 2: In the Discussion, the reference to statistical evaluation is inappropriate because a statistical analysis has not been performed; the authors reported only the percentage of the cases in which concordance and discrepancy, respectively, were observed.

Response: Thank you for pointing this out. Analysis was performed to assess the correlation between the autopsy and CT scan results. The authors have added sensitivity (percentage of those with injury that have injury detected on CT scan), and positive predictive value (percentage of positive CT results that have injury) for the CT scan based on the autopsy results, for the each type of injury. The data were incorporated to our manuscript.

Comment 3: The authors should make some considerations about the influence of radiologist skill and expertise on CT images analysis. This is important considering that the image revision has not been blindly performed.

Response: We agree with this comment. We have added the details about the radiologists’ skills to our article. CT images were reviewed by experienced clinical certified radiologists. In the first step the image revision has been performed blindly, and radiologists made their decisions independently.

Comment 4: The Conclusions are good. However, the Authors should better express the concept at line 458 “Unless the diagnosis by CT is 100% ...”

Response: Agree. We have, accordingly modified that sentence to emphasize this point. We have highlighted the changes within the manuscript.

Reviewer 2 Report

The authors conducted a retrospective-prospective study involving 510 traumatic deaths, and compared selected traumatic findings through the last antemortem CT examination and autopsy.

The topic is interesting but some points have to be clarifyed.
- The kind of antemortem CT: always whole body? Mirate scan?
- The authors said that “CT scan was performed within 48 hours before death (except for fractures)”: if so, in case of repeated examinations they used the last one within 48 hs before death.

Why “except for fractures”? The time lapse was longer than the fractures and when?
If so (CT scan > 48 hrs in fractures ), the longer interval could have played a role in limited diagnosis of some fractures or cranial hemorrhage.
- The authors said that “More than a quarter of cases (26.8%) were deaths that occurred within 24 hours of admission to the medical facility”: was CT scan always performed in all cases of brief hospitalization ?
- The authors compared antemortem CT findings using the medical reports and, only in discrepancy, by reviewing of CT images. Maybe the primary use of CT medical reports instead of the images could affect the results, it could be better the use of CT images, always. Was this option ever considered by the authors?
- Why was the premortem CT scan performed in the hanging cases (during hospitalization, I suppose)?

- A statistic method should be applied to remark the significance level.

Author Response

Comments from Reviewer 2

Comment 1: The kind of antemortem CT: always whole body? Mirate scan?

Response: In the group of traffic accidents and falls from a height, a whole-body CT was always performed. In other cases, CT was performed with the focus on the injured body area (head, head + neck, torso). The authors have specifically added and highlighted all the necessary details within the manuscript.

Comment 2: The authors said that “CT scan was performed within 48 hours before death (except for fractures)”: if so, in case of repeated examinations they used the last one within 48 hs before death.

Response: In our study, in case of repeated CT examinations, we always used the last one performed before death (details were added and highlighted in the article).

Comment 3: a) Why “except for fractures”? The time lapse was longer than the fractures and when?
b) If so (CT scan > 48 hrs in fractures), the longer interval could have played a role in limited diagnosis of some fractures or cranial hemorrhage.

Response: a) Cases were included in the study if an autopsy was conducted within 48 hours after death, a CT scan was performed within 48 hours before death, in cases of fractures within 96 hours before death, and there was no surgical intervention. Another criterion was a sufficient radiologic and morphological interpretation of the traumatic findings necessary for comparison.

  1. b) Thank you for pointing this out. Precisely because the longer interval can play a role in limited diagnosis of some fractures or cranial hemorrhage, cases of fractures were included in the study if a CT scan was performed within 96 hours before death. The details were added and highlighted within the manuscript.

Comment 4: The authors said that “More than a quarter of cases (26.8%) were deaths that occurred within 24 hours of admission to the medical facility”: was CT scan always performed in all cases of brief hospitalization?

Response: In all cases CT scan was always performed.

Comment 5: The authors compared antemortem CT findings using the medical reports and, only in discrepancy, by reviewing of CT images. Maybe the primary use of CT medical reports instead of the images could affect the results, it could be better the use of CT images, always. Was this option ever considered by the authors?

Response: Thank you for this suggestion. However, in the case of our study, the authors compared CT images only in cases of discrepancy. CT images were evaluated by experienced clinical certified radiologists, who have made their decisions independently. In cases in which our findings coincided with the CT findings described in the medical reports, we did not consider use of CT images necessary (we had no reason to compare CT images).

Comment 6: Why was the premortem CT scan performed in the hanging cases (during hospitalization, I suppose)?

Response: In all evaluated cases of hanging, a CT scan was always performed during the patient´s hospitalization. In our conditions (in our hospitals), a CT examination is performed as a standard procedure in these cases.

Comment 7: A statistic method should be applied to remark the significance level.

Response: Thank you for the suggestion. The authors have added to the article sensitivity (percentage of those with injury that have injury detected on CT scan), and positive predictive value (percentage of positive CT results that have injury) for the CT scan based on the autopsy results, for the each type of injury. The data were incorporated to our manuscript.

Due to the fact that the authors evaluated only positive findings (injuries), it was not possible to calculate the specificity and negative predictive value. It was also not possible to use several statistical methods, particularly the chi-square test or Student´s t-test. Please, if the reviewer knows some statistical method that could be used, give us advice, and we will use it and add it to the article.

Round 2

Reviewer 2 Report

The Authors have provided adequate responses to the remarks.

Author Response

We appreciate the time and effort that you have dedicated to providing your valuable feedback on our manuscript. We are grateful for your insightful comments on our paper. We are happy that we have been able to incorporate changes to reflect all your suggestions. Thank you.